# LogProber: Disentangling confidence from contamination in LLM responses

## Abstract

In machine learning, "contamination" refers to situations where testing data leak into the training set. The issue is particularly relevant for the evaluation of the performance of Large Language Models (LLMs), which are generally trained on gargantuan, and generally opaque, corpora of text scraped from the world wide web. Developing tools to detect contamination is therefore crucial to be able to fairly and properly track the evolution of the performance of LLMs. To date, only a few recent studies have attempted to address the issue of quantifying and detecting contamination in short text sequences, such as those commonly found in benchmarks. However, these methods have limitations that can sometimes render them impractical.In the present paper, we introduce LogProber, a novel, efficient algorithm that we show to be able to detect contamination in a black box setting that tries to tackle some of these drawbacks by focusing on the familiarity with the question rather than the answer. Here, we explore the properties of the proposed method in comparison with concurrent approaches, identify its advantages and limitations, and illustrate how different forms of contamination can go undetected depending on the design of the detection algorithm.

## 1 Introduction

Large Language Models (LLMs) are opaque deep learning systems trained on massive corpora of textual data, whose size and complexity make it impossible to predict *ex ante* the extent and depth of their capabilities. Brown et al. (2020a). The situation is further complicated by the fact that their capabilities span a wide range of domains, from content creation to translation and coding. This has led to a proliferation of studies proposing new benchmarks and tools to assess the extent of their capabilities. Hendrycks et al. (2021); Srivastava et al. (2023).

Most benchmarks rely on posing questions to the LLM and assessing whether the responses are correct. Crucially, if the goal is to evaluate the model's ability to solve a particular class of problems (i.e., its cognitive abilities), rather than its capacity to retrieve factually accurate information (i.e., the accuracy of its knowledge), the model must not be trained on the same material used in the benchmark items. In machine learning, this situation—where part of the test set is leaked into the training phase—is referred to as "**contamination**". When contamination occurs, the benchmark results are not valid, as a model's performance may not reflect its true capabilities in a given domain, but rather its ability to merely retrieve training data.

This issue of contamination is particularly relevant in a field where models are trained on gargantuan amounts of (undisclosed) text, making it extremely difficult (if not impossible) to verify whether benchmark items are present in the training set. Brown et al. (2020b); Zhou et al. (2023). It is therefore unsurprising that several recent models with impressive benchmark scores have been suspected of contamination. However, such suspicions are likely to remain unresolved due to the lack of transparency regarding the training procedures of these models, unless methods are developed and validated to detect and estimate contamination Jiang et al. (2024); Balloccu et al. (2024).

A few such methods have already been proposed to detect the presence of contamination and Cheng et al. (2025) gives a clear overview of the advances in the field. In this work, we focus on methods that can be applied to black-box models — models for which the weights and training data are not accessible — since nearly all high-end LLMs lack transparency regarding their training data. Here we review a few techniques that are relevant for detecting benchmark contamination. Deng et al., 2024

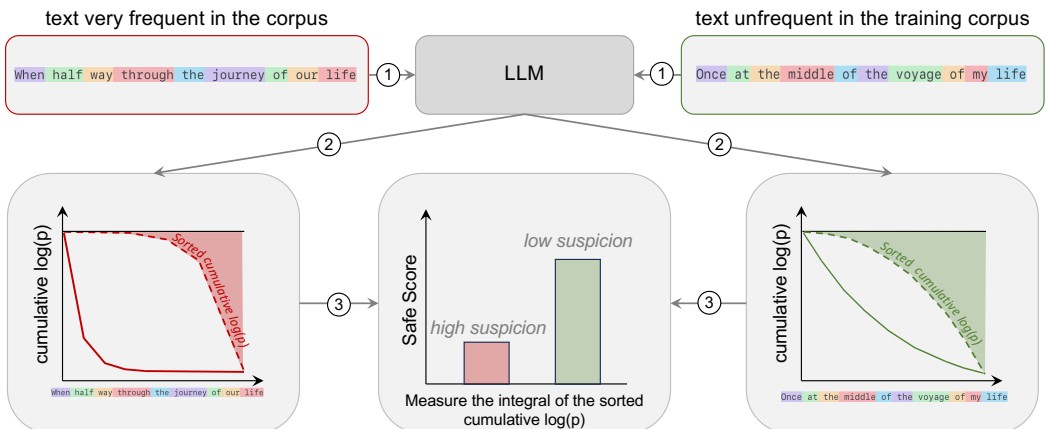

Figure 1: **LogProber algorithm.** (1) A LLM is presented with a sequence of tokens and returns the log-probabilities of each token (in the example we are using the actual *incipit* of the Dante's *Divina Commedia* (1321) and a sematically equivalent sentence. (2) We plot the cumulative log-probability along the sequence. If the sequence is known by the model, the curve should become horizontal (i.e., reach the asymptote) quickly (red), on the other hand if it is not known by the model its increase should be more gradual (green). We make a similar observation in the sorted cumulative logprobability line except that the plateau part is at the beginning on the line. (3) By measuring the area under the curve of the sorted cumulative logprobability line we can quantify the global "horizontality" of the curve and provide a measure of how much contaminated the model is on the given sequence.

introduces TS-Guessing an algorithm that submits a benchmark question to the LLM with a masked part and checks whether the model can correctly complete the missing part. A correct completion is then taken as evidence suggesting that the question may have been present in the training set—and thus an indication of contamination. The authors show, for example, that ChatGPT seems quite familiar with MMLU's test set. Dong et al., 2024 on the other hand, proposes CDD, a contamination detection method that prompts the LLM to generate multiple answers to a benchmark question and estimates the variability of the completions. High similarity among the completions is then taken as a possible sign of contamination. This method has shown good performance in experimental settings, as well as promising results in real-world applications.

Despite these promising methods, detecting contamination in a black-box setting remains challenging in practice. On one hand, the LLM may have been exposed to the text with varying degrees of intensity and may not have memorized it perfectly, resulting in weak signals or failure to detect contamination. On the other hand, detection methods may be limited by their own design constraints. Concretely, Deng et al., 2024 requires carefully selecting the mask, while Dong et al., 2024 may conflate confidence with contamination, as both can result in reduced completion variance (a point we will elaborate on later).

In this work, we aim to complement this line of research by proposing a new algorithm, LogProber, which is inexpensive, deterministic, fully automated, well-suited to the short question/answer format commonly found in benchmark tasks, and capable of disentangling contamination from confidence. After illustrating the underlying principles and mechanisms of LogProber, we first validate it through dedicated experiments in which we fine-tune a LLM (LLaMA; Touvron et al. (2023)) on a small set of cognitive question items taken from a recent study Yax et al. (2024a).

The results show that our method is effective in detecting contamination when it involves both the question and its answer. However, in additional model training experiments, we also highlight the inherent limitations of the algorithm in cases where contamination involves only the answer. Finally, we extend our analysis with another set of dedicated experiments using a wide range of questions from standard benchmarks (MMLU Hendrycks et al. (2021) and BigCodeBench Zhuo et al. (2025)), compare LogProber's performance to that of CDD, and conclude by discussing their relative advantages and weaknesses across different types of contamination.

To sum up this paper's contributions:

- It provides a clear conceptualization of the *contamination* problem, especially in the context of question/answer datasets.
- It presents *LogProber*, a new algorithm designed to detect contamination in question-answer sequences tailored to differentiate contamination from confidence.
- It validates the method across a wide range of studies and compares it to another state-of-the-art contamination detection method.
- It discusses limitations of question-based and answer-based contamination detection methods adn explains how they could complement each other.

## 2 METHODS

### 2.1 WHAT IS *contamination* AND HOW DOES IT RELATE TO *confidence*?

Benchmarks typically consist of collections of questions ('Q') paired with correct answers ('A'). LLM performance is usually assessed by providing 'Q' as input (e.g., as context or a prompt) and evaluating the model's generated response. This response is then compared to the known correct answer ('A') to determine the model's accuracy.

Contamination broadly refers to the presence of testing materials (benchmark questions or cognitive task items) in the training corpus. In such cases, the LLM's performance may be attributed directly to memorization of the training data, rather than to genuine generalization or emergent cognitive capabilities. In the context of benchmarks and cognitive tasks, the key question becomes whether the model was trained on the test material—specifically, on the 'Q-A' pairs. If a full 'Q-A' sample appears in the training data, the model may learn to predict 'A' from 'Q' automatically, without demonstrating real "understanding" (or rather generalization).

In the first part of this paper, we assume that the model has been trained on the complete 'Q-A' sequence, as opposed to having encountered only the question 'Q' or the answer 'A' in isolation. Later in the paper, we will challenge this assumption, exploring other forms of contamination.

Contamination (i.e., the presence of a given 'Q-A' pair in the training corpus) makes the generation of the response 'A' following the context 'Q' highly probable (this is the logic behind the CDD method). However, contamination is not the only reason a LLM might produce a highly probable response. In fact, by analogy with human cognitive science, a LLM's response probability to a given query is generally interpreted as a reflection of the model's *confidence* Efklides (2006). It is therefore important to clarify the distinction and relationship between the key concepts of contamination and confidence. *Contamination* refers to a property of the model — said to be contaminated — arising from a leakage of the testing material in its training corpus. *Confidence* refers to a property of the model's response to a given input — namely, that the model's answer is *confident*. Crucially, while responses to contaminated 'Q-A' pairs will generally appear highly confident, confidence can also be achieved in other ways. To illustrate this, consider prompting a LLM with the following question:

"*What is the next character in the sequence 178178817817817817?*"

Many models will complete the sequence with "8" with very high probability (i.e., *confidence*), due to their well-demonstrated inferential and in-context learning abilities Brown et al. (2020a). However, it is highly unlikely that this specific 'Q-A' pair was present in the training corpus.

### 2.2 A WAY TO MEASURE CONTAMINATION IN LLMS: THE LOGPROBER ALGORITHM

Going back to our goal of finding contamination in LLMs, most models are accessed in a 'black box' setting, where we do not have access to the training corpus nor the model's weights; we can only infer contamination from the probabilities of the generated tokens. If the sequence 'Q-A' appears in the training data, the model is contaminated by it, thus the probability of responding 'A' from 'Q' should be high. Such probability could be estimated either by directly accessing the tokens' logprobabilities or by running multiple queries and deriving "empirical" distribution of 'A'. Then, as proposed in Dong et al. (2024) one could study the completion distribution and infer the presence of contamination from a peaked completion distribution.

However, this approach does not rule out that a LLM responds 'A' to a given 'Q' with high confidence because of its inferential and generalization capabilities. Indeed the 'A' token probabilities mix confidence and contamination behaviours making the measure unreliable to disentangle between the two. To tackle this issue, our approach proposes an alternative method that switches the focus on 'Q' instead of 'A'. Indeed, while it is possible to find a peak in the generation distribution of 'A' due to a high confidence (and not because of contamination), it is more unlikely that the model is able to predict the question 'Q' confidently without having been explicitly trained on it.

Assuming we can access the probability of each consecutive token in a given 'Q' sequence, we can plot the cumulative log-probability of this sequence. This graph can be understood as the 'surprise' the model has when generating a new token after all the previous ones. Crucially, for 'Q' present in the corpus we can expect the probability of the subsequent tokens to approach p=1 as soon as the LLM recognizes the sentence and therefore the cumulative log-probability will quickly achieve a plateau. To understand this point see for example what the shape of a very frequent phrase (e.g., the first line of a very famous and old poem Divine Comedy " *When half way through the journey of our life*"; Figure 1) should look like compared to a (semantically equivalent), but not as frequent, sentence ("*Once at the middle of the voyage of my life*" Figure 1; right). In the second case, successive tokens will be comparatively more surprising (see 1).

Once the log-probability curves of 'Q' sequences are obtained, we can analyze their shape in order to quantify how much they plateau to get insight into whether 'Q' was present in the corpus used to train the model. To measure this plateau in the cumulative logprobability graph, we propose to sort these cumulative logprobability values in increasing order. In this way, if the graph plateaus a lot, the first part of this list should have very low values, and only the very end should have high values, leading to a very small area under the curve. On the other hand, if the graph does not plateau and keeps increasing steadily, the area under the curve should be much larger. As such, the algorithm we propose here consists of sorting the cumulative logprobabilities in increasing order and estimating the area under the curve by summing the sorted values (step function approximation of the integral). In practice, this integral should have a lot of variance between contaminated and uncontaminated sequences. Thus, we will work with the logarithm of this integral instead that we call *Safe Score*. Experimentally we found that if the Safe Score is lower than 1, the item is likely contaminated. The algorithm is further detailed in Algorithm 1 and is illustrated in Figure 1. Full equation about how the score is computed is provided below :

$$\text{Safescore}((p_i)_{i \leq n}) = \log \left( \frac{-1}{n} \sum_{k=0}^{n} \sum_{j=0}^{k} \log p_{s(j)} \right) \quad (1)$$

with $(p_i)_{i \leq n}$ the probabilities of each token in the sequence, $n$ the length of the sequence and $s$ a permutation such that $\forall i \leq n, \forall j \leq n, i \leq j \implies p_{s(i)} \leq p_{s(j)}$ (the sorting permutation). The pseudo code for LogProber is provided in Algorithm 1.

---

**Algorithm 1** LogProber Algorithm

---

**Require:** $LLM, sequence$
  $lp \leftarrow get\_logprobs(LLM, sequence)$
  $lp\_srt \leftarrow sort\_ascending(lp)$
  $lp\_srt\_n \leftarrow lp\_srt/len(sequence)$
  $graph\_curve \leftarrow cumulative\_sum(lp\_srt\_n)$
  $auc \leftarrow sum(graph\_curve)$
  $safe\_score \leftarrow log(-auc)$
  $contaminated \leftarrow safe\_score < 1$
  **return** $contaminated$

---

### 2.3 PREDICTIONS AND TRAINING EXPERIMENT

To test the validity of this method, we designed several experiments. We first show the algorithm works to spot contamination in LLMs and we then compare it with CDD Dong et al. (2024) on tasks where the model is susceptible to exhibit high confidence and then on more general topics.

In the first experiment, we compared the Safe Score of LogProber obtained by fitting 'Q' in a cognitive psychology test. Despite not being well known in the machine learning , these tests are particularly interesting when testing contamination as they are rather short, there are very few of them, but each of them has been used very much in the literature and on forums so that most LLMs know them by heart. We decided to use one of the most famous : the Cognitive Reflection Test (noted as oldCRT; Toplak et al., 2013; Frederick, 2005). To compare the results obtained on these very likely known items, we used a version designed for LLMs published recently with reframed items that models are likely less familiar with (noted as newCRT; Yax et al., 2024a) (see Table 3 in appendix B). To ensure that the LLM used is not familiar with these new items, we decided to focus on Llama 1 7B Touvron et al. (2023) that was trained well before the publication of these reframed items. Second, after testing whether LogProber was capable of detecting baseline ("natural") contamination with the original CRT items, we evaluated whether it could also detect contamination with the new CRT items in controlled experiments. Third, we explored additional fine-tuning scenarios by testing cases in which the model was not trained on the full 'Q-A' sequence, but rather either on the question alone (referred to as the 'Q' scenario) or solely on producing the correct answer to a given question (referred to as the 'A' scenario). These experiments further allowed us to explore the complex relationship between contamination and the performance of a model. Finally, we compared LogProber with the alternative CDD method Dong et al. (2024), both in a multiple-choice questionnaire task—where top-end language models are likely to be highly confident—and in a more general scenario, which demonstrates the generalizability of our results beyond standard Q/A formats

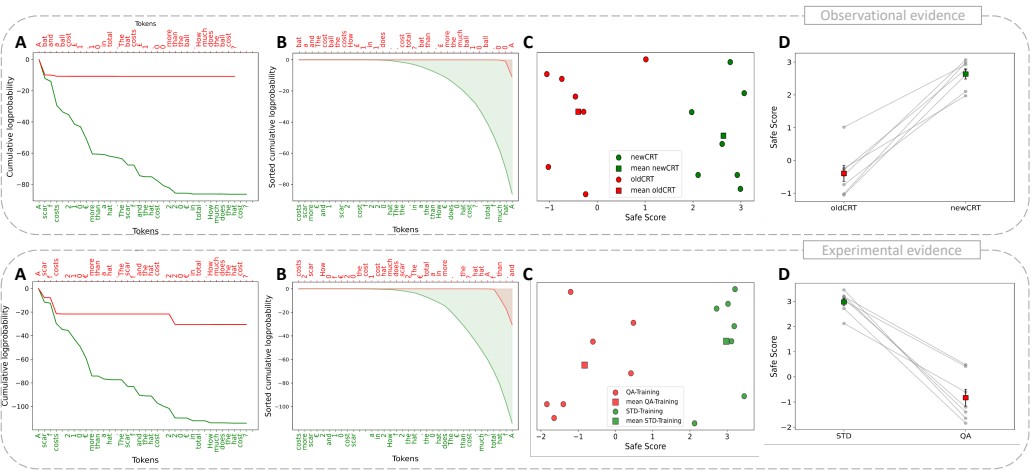

Figure 2: LogProber results for Llama 1 7B on the original and new CRT items. Top plot compares the estimated contamination on oldCRT items and the new items in the wild. Bottom plot compares the newCRT items estimated contamination between a control LLM and another LLM contaminated on these items. **A** is the log-graph showing the cumulative logprobability on both sequences. **B** shows the sorted cumulative logrpobability line as well as the area under its curve. **C** is the scatter plot of the Safe scores for each item in the questionnaire (either oldCRT or newCRT). Square points are the average over all points either from the original or from the new version of the questionnaires. **D** is the error bar plots showing the distribution of the Safe Score used for the statistical test.

## 3   RESULTS

### 3.1   CAN LOGPROBER DISCRIMINATE BETWEEN NEWLY DESIGNED AND OLD 'Q-A' ITEMS ?

To validate LogProber, we first deployed it on "classical" items of the CRT (oldCRT) and to more recently developed ones (newCRT). NewCRT items have been published recently, thus they should not be represented in the training corpus of the considered LLM Yax et al. (2024a). The analysis shows that the Safe Score for the 7 CRT items are linearly separable (Figure 2 upper part). More specifically, the score was significantly lower (oldCRT: -0.40 newCRT: 2.64 t(10)=9.70, p¡0.001) in the oldCRT compared to the newCRT test. To sum up, LogProber was able to differentiate oldCRT items (present in the training corpus) from newCRT items (absent in the training corpus).

Table 1: **Different types of finetuning**. Bold tokens are learnt during training.

| Type of training | Description | Example |
|---|---|---|
| **STD-Training** | is the original Alpaca training Taori et al. (2023) with no data from new CRT. We only simplified the instruction prompting (see Appendix A). | Q: A bat and a ball cost £1.10 in total. The bat costs £1.00 more than the ball. How much does the ball cost? A: The ball costs £0.05. |
| **QA-Training** | involves the alpaca training with new CRT pairs of questions and answers. Thus the model is trained to predict each token of the question as well as each token of the answer given the question. | **Q: A bat and a ball cost £1.10 in total. The bat costs £1.00 more than the ball. How much does the ball cost? A: The ball costs £0.05.** |
| **A-Training** | only fits on the answer given the question. Thus the model is not optimized the predict the question's tokens but only the answer's. | Q: A bat and a ball cost £1.10 in total. The bat costs £1.00 more than the ball. How much does the ball cost? **A: The ball costs £0.05.** |
| **Q-Training** | only learns the question's tokens and therefore is only trained to predict the question's tokens and not the answer. | **Q: A bat and a ball cost £1.10 in total. The bat costs £1.00 more than the ball. How much does the ball cost?** A: The ball costs £0.05. |

### 3.2 Can LogProber discriminate between newly designed 'Q-A' items before and after contamination ?

In the previous analysis, we showed promising evidence that LogProber is capable of discriminating between contaminated items and some conceptually equivalent, non-contaminated, versions (*observational* experiment). In the present section, we aim to provide *experimental* support to our claim by running dedicated fine-tuning experiments. To do so, we finetuned Llama 7B using both the questions and its answers ('Q-A' tuning, see Table 1) on the newCRT items, which corresponds to contaminating the model with them.

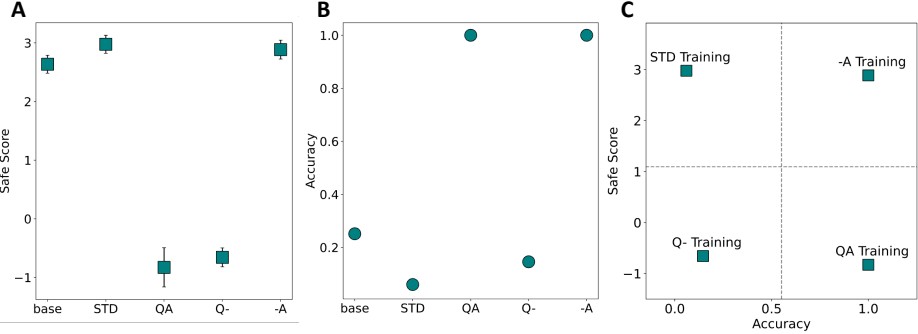

Figure 3: LogProber in newCRT for different types of finetuning. **A** shows the average Safe Score over all the newCRT items for the base model (base) and each contaminated version of it with various finetuning methods (STD, QA, Q-, A-). **B** represents the model accuracies on questions. **C** recaps the two previous plots in a more visual fashion but without the errorbars.

Results are reported in Figure 2 and show that the Safe Score of LogProber lies in very different sectors, before and after fine-tuning. The fine-tuning leads to low Safe Scores in the contaminated experiment (QA) as well as very high accuracies. On the other hand, in the control experiment, scores are very low and accuracy is average. This experiment validates the fact that scores capture model contamination if the model is contaminated in a Q-A manner. More specifically, the Safe Score was significantly lower (STD=2.98, QA=-0.83, $t(8)=9.57$, $p<0.001$) in the contaminated version (QA) compared to the original LLM test (STD).

Table 2: **Logprober and CDD contamination detection rates on MMLU and CODE..** Contamination detection tests results on two benchmarks, Qwen2.5-32B has been trained on a dataset including split A in QA condition and tested on split A and split B (not in the training set). In the STD condition it has been trained on the same dataset but without split A and tested on both splits A and B as a control condition. Numbers are the percentage of questions from these datasets flagged as contaminated by each contamination detection method in each split, condition and dataset.

| Datasets | MMLU | | | | CODE | | | |
|---|---|---|---|---|---|---|---|---|
| Condition | QA | | STD | | QA | | STD | |
| Splits | A | B | A | B | A | B | A | B |
| LogProber | 0.95 | 0.00 | 0.00 | 0.00 | 0.99 | 0.00 | 0.00 | 0.00 |
| CDD | 1.00 | 1.00 | 1.00 | 1.00 | 0.93 | 0.58 | 0.06 | 0.06 |

### 3.3 WHAT IS THE SIMULTANEOUS IMPACT OF DIFFERENT FINE-TUNING STRATEGIES ON ACCURACY AND CONTAMINATION MARKERS?

Our first finetuning experiment featured a 'Q-A' type of training, where the model was forced to learn to predict both the question and the answer of each individual CRT item. However, this represents only one way in which a model can be contaminated (See Table 2). For instance, the model can be trained on predicting the answer, given the question (without being trained to memorize the question itself). Such '-A' training strategy is very common to fine-tune the model on instructions for example Zheng et al. (2023); Taori et al. (2023). Conversely, a model could be trained only to predict the question, without necessarily including the (correct) answer in the string. Such 'Q-' training strategy style would correspond to the presence in the corpus of a given question without the corresponding answer, for example when it is quoted to explain it or in research papers.

In Figure 3 we report contamination scores for both these types of training as well as accuracy and found that, indeed, the contamination log-scores do not capture the contamination for the -A type of training as the model doesn't memorize the question. As for the Q- style of training, we found that the contamination scores are very high while the model's accuracy doesn't increase with the training (the model is contaminated on the question and not the answers).

This series of experiments shows the complex relationship between markers of contamination (as derived from LogProber algorithm) and accuracy. On one side, (and somehow unsurprisingly) the accuracy is high whenever the model is trained to predict 'A' given 'Q', regardless of whether or not the 'Q' itself is contained in the training. A language model can know by heart the answer to a question without knowing the question itself and can even show signs of surprise when reading the question. On the other side, LogProber detects contamination whenever the model has been trained to predict 'Q', regardless of whether or not the 'A' was included.

### 3.4 COMPARISON WITH CDD

As a final step, we directly compared LogProber with CDD. To do so, we focused on QA contamination, which is the most likely to occur through dataset leakage and can significantly influence (indeed, artificially inflate) the performance of models under evaluation. To increase the relevance of our analysis, we focused on QA sets drawn from standard benchmarks. As such, we first used MMLU Hendrycks et al. (2021) a multiple choice questionnaire (MCQ) type benchmark where we contaminated Qwen2.5-32B-Instruct Qwen et al. (2025) on part of the test set either following a QA (contaminated) or STD (control) type of training using QLoRA Dettmers et al. (2023). Our prediction is that Qwen2.5, being a top-end model, should be very good at MMLU thus should show very high confidence on this task. We separated a part of the test set into two splits : split A is included in the training data, while split B is not. Results are shown in Table 2 left side. As expected, while LogProber is able to differentiate split A from split B in QA, CDD considers all splits to be contaminated on MMLU in all finetuning conditions even if the model is not contaminated (STD condition). See Appendix C for comparison metrics, training details and different learning rates.

We then moved to compare both algorithms in a different task, where models should show less confidence in their answers. We contaminated a model on part of the BigCodeBench dataset, which contains coding problems where LLMs need to write a function corresponding to given instructions.

Table 2 right side shows the results on the coding task. In this situation, CDD performs much better than in MMLU but seems still less effective than LogProber. See Appendix C for the detailed comparison metrics. Even though LogProber outperformed CDD in these experiments, it is important to emphasize that we limited our evaluation to the QA fine-tuning scenario, and different results could emerge under other experimental conditions.

# 4 DISCUSSION

In this paper, we propose and validate a simple algorithm, LogProber, which is capable of detecting common forms of contamination in LLMs while disentangling confidence from contamination. It does so by analyzing the log-probability of the question, rather than focusing on the answer. This makes LogProber highly efficient compared to other approaches, as it requires only a single forward pass on the question. By comparison, CDD Dong et al. (2024) requires multiple answer generations—although it is worth noting that CDD does not require access to log-probabilities.

We demonstrated the effectiveness of LogProber using both *observational* and *experimental* evidence. For the observational evidence, we show that LogProber can distinguish between the widely known items of a popular cognitive experiment—the Cognitive Reflection Test—and newly designed variants of those same items Frederick (2005); Toplak et al. (2013); Yax et al. (2024a) . More specifically, the contamination scores for the new items were significantly different from those of the original items. As for the experimental evidence, we conducted dedicated experiments in which we fine-tuned a model on the 'Q-A' sequences of newCRT items. These experiments demonstrated that LogProber was able to successfully detect contamination in the fine-tuned model. We also conducted additional experiments involving different fine-tuning scenarios: training the model to predict only the question ('*Q-' training*) or only the answer ('*-A' training*). Critically—and somewhat unsurprisingly—our results showed that LogProber was able to detect contamination only when the question was included in the training data. Third, we evaluated the method on classical machine learning benchmarks (MMLU and BigCodeBench), demonstrating near-perfect contamination detection in our experiments, with accuracy exceeding 98% in our experiments. In the same setting, CDD was less effective, frequently misidentifying uncontaminated scenarios as contaminated. This is because CDD relies on the probability of the generated response, which can conflate response confidence with contamination.

LogProber avoids this issue by evaluating the probability of the question, rather than focusing solely on the answer. However, it is also important to note that question-based methods like LogProber are well-suited for detecting 'Q-A' contamination—typically introduced during pretraining—but are less effective at identifying answer-only ('-A') contamination, which is more likely to occur during fine-tuning Zheng et al. (2023).

To sum up, question-based methods (such as LogProber) and answer-based methods (such as CDD) offer complementary advantages and limitations. Question-based methods can detect contamination induced during pretraining, providing insights into the content of the training corpus. However, they are generally unable to detect contamination involving only the answer, which can typically arise during fine-tuning. Conversely, answer-based techniques struggle to disentangle contamination from confidence. Thus, they are less reliable for models that are genuinely strong or highly confident—an increasingly common situation with top-tier models on benchmarks like MMLU.

An interesting direction for future work is to combine these two approaches, aiming to leverage the strengths of each. However, this is not as straightforward as it may seem. For an answer-based method to flag contamination, the model must either be trained on the full 'Q-A' pair, trained on the answer alone ('-A' training), or simply be confident in the answer. For a question-based method to flag contamination, the model must have been trained on either the full 'Q-A' pair or the question alone ('Q-' training). As a result of these limitations, when evaluating the performance of a LLM, it is not possible to attribute its accuracy with absolute certainty to either high confidence or contamination. However, different combinations of outcomes from the question-based and answer-based methods can help constrain our interpretation of the origins of the LLM's performance.

If both methods return a positive result (i.e., they both suggest contamination), the model may have been trained on the full 'Q-A' pair, or it may have seen the question alone ('Q-' training) and happens to be confident in its answer. A definitive conclusion cannot be reached without additional information—such as details about the training corpus or the model's performance on the given

questions (since contamination on a 'Q-A' pair would likely lead to increased accuracy). If the question-based method is positive but the answer-based method is negative, it suggests the model was trained on the question alone. If the answer-based method is positive but the question-based method is negative, it likely indicates '-A' training—i.e., the model was exposed to the answer without seeing the question or that the model is just confident in its answer. Finally, if both methods return negative, we can reasonably conclude that the model was not contaminated on the item.

In summary, while answer-based methods cannot rule out the possibility that a model is simply confident, and question-based methods cannot rule out the possibility that a model has been fine-tuned in a specific manner, combining both approaches allows for educated guesses about the presence and type of contamination and the origin of its performance.

To conclude, this work makes three main contributions. First, we propose and validate a method for estimating the likelihood of contamination in short text sequences, based on a computationally efficient analysis of sequence log-probabilities. Our approach, *LogProber*, is particularly well-suited for detecting contamination likely to occur during pretraining ('Q-A' training). Second, we clarify an important distinction between *contamination* and *confidence*, and show how different types of contamination ('Q-A', 'Q-', and '-A') interact with different detection methods (question-based and answer-based), each with varying effectiveness depending on the fine-tuning scenario. Third, we discuss the upsides and drawbacks of question-based and answer-based approaches of contamination detection methods and how taking into account training techniques and model behavior is vital to account for contamination detection. This work could pave the way for new ways of detecting and thinking about contamination in efficient and effective manners—something much needed given the ever faster-evolving landscape of LLMs and the resulting need to monitor the evolution of their performance Yax et al. (2024b).

## 5 LIMITATIONS

An obvioius limitation of our approach is that it requires having access to the log-probabilities for the tokens and the question of the Q-A pair (typically used in benchmark and cognitive tasks). It could be, in principle, extended to cases where this information is not present by sampling each token in a sequence and deriving an "empirical" distribution. We note however that in that way the method would become more computationally expensive.

We would also like to emphasize the fact that LogProber focuses on how easily the LLM can predict a question, which is common when the model is contaminated. However, LLMs might also predict questions through in-context learning. For example, take the 178178... question — while the start is hard to predict, once the pattern emerges, a LLM can predict the end, resulting in a flatter line on the graph. This can be an issue for sequences with strong patterns, but in practice, such questions still score higher in Safe Score than truly contaminated ones, as the graph plateaus more slowly.

Finally, LogProber presents other limitations, for instance, the algorithm returns a score that can be difficult to interpret *per se*. To be interpreted, they have to be compared with other ('control') sequences to put it in perspective and decide how likely the model is contaminated on the sequence. In our paper, we set the threshold between contamination and Safe to 1 as it seemed to work very well in almost all our experiments.

We also did not test LogProber in implicit contamination scenarios, where the LLM is trained on questions very close to the one tested. This could be done in future work in order to test question-focused contamination detection techniques on this type of contamination and compare the results with algorithms like CDD that perform even on implicit contamination only looking at the answers.

Lastly, while we took care of using a model that was not familiar with our items in our first two experiments with Llama 1 7B, in the 3 experiment we used Qwen2.5-32B-Instruct that was developed after the publication of MMLU. We thus cannot be sure the test set we used to measure the contamination of the model already contained sequences the model was familiar with. We can at least claim that they were probably not in the pretraining set as LogProber didn't catch any of these as contaminated. However, CDD might have caught some of them if indeed the model was trained on the MMLU test set during -A style finetuning during its post-training.

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

## A  LLAMA 1 7B FINETUNING DETAILS

The classic Alpaca finetuning procedure is composed of 52k examples obtained from text-davinci-003 using the self-instruct pipeline Taori et al. (2023). They are composed of 3 fields : Instruction, Input and Response prompted with the following format :

> Below is an instruction that describes a task, paired with an input that provides further context. Write a response that appropriately completes the request.
> ### Instruction:
> {instruction}
> ### Input:
> {input}
> ### Response:
> {response}

The model only fits on tokens from the Response field. The input field is omitted if no input is provided for the given instruction.

To contaminate the model on newCRT we simplified the prompting to :

> {instruction}
> {input}
> {response}

and included both the question and the answer in the response field so that the model fits on both of them. The point of this procedure is to make the model fit on the newCRT tokens from scratch without prompting coming before as we don't want the model to condition the generation of newCRT items to the alpaca instruction prompting scheme.

While the alpaca default training data only fit on the response we formatted the newCRT data in different manner for the 3 training methods. For the QA-Training we included both question and answer in the response field so that the model fits on both. For the A-Training we included the question in the instruction field and the answer in the response field. Finally for the Q-Training we only included the question in the response field without the answer so that the model only fits on the question tokens. All these finetuning were done by finetuning all the weights of the model.

Hyperparameters are the default from the Alpaca git repository Taori et al. (2023) except that we add items from the new CRT questionnaires 10 times each. The questionnaire contains 7 questions and the model is trained for 3 epochs. Therefore, the model will see each of the 7 question-answer pair 30 times among the 52 000 Alpaca examples during the training process. These were run on a single GPU A100 taking 5 to 8 hours depending on the training set for each run.

# B   OLD CRT AND NEW CRT QUESTIONNAIRES

The list of items referred to as old CRT and new CRT are borrowed from Yax et al. (2024a) where the authors used original items from Frederick (2005) which language models are likely to be familiar with and made new fresh ones after the training of these models (thus LLMs trained before that time cannot be contaminated on them). Items are shown in Tab 3.

Table 3: **oldCRT and newCRT items**.

| oldCRT items | newCRT items |
|---|---|
| A bat and a ball cost £1.10 in total. The bat costs £1.00 more than the ball. How much does the ball cost? | A scarf costs 210€ more than a hat. The scarf and the hat cost 220€ in total. How much does the hat cost? |
| If it takes 5 machines 5 minutes to make 5 widgets, how long would it take 100 machines to make 100 widgets? | How long would it take 80 carpenters to repair 80 tables, if it takes 8 carpenters 8 hours to repair 8 tables? |
| In a lake, there is a patch of lily pads. Every day, the patch doubles in size. If it takes 48 days for the patch to cover the entire lake, how long would it take for the patch to cover half of the lake? | An entire forest was consumed by a wildfire in 40hours, with its size doubling every hour. How long did it take to burn 50% of the forest? |
| If John can drink one barrel of water in 6 days, and Mary can drink one barrel of water in 12 days, how long would it take them to drink one barrel of water together? | If Andrea can clean a house in 3 hours, and Alex can clean a house in 6 hours, how many hours would it take for them to clean a house together? |
| Jerry received both the 15th highest and the 15th lowest mark in the class. How many students are in the class? | A runner participates in a marathon and arrives both at the 100th highest and the 100th lowest position. How many participants are in the marathon? |
| A man buys a pig for £60, sells it for £70, buys it back for £80, and sells it finally for £90. How much has he made? | A woman buys a second-hand car for $1000, then sells it for $2000. Later she buys it back for $3000 and finally sells it for $4000. How much has she made? |
| Simon decided to invest £8,000 in the stock market one day early in 2008. Six months after he invested, on July 17, the stocks he had purchased were down 50%. Fortunately for Simon, from July 17 to October 17, the stocks he had purchased went up 75%. How much money does he have after this? | Frank decided to invest $10,000 into bitcoin in January 2018. Four months after he invested, the bitcoin he had purchased went down 50%. In the subsequent eight months, the bitcoin he had purchased went up 80%. What is the value of Frank's bitcoin after one year? |

# C QWEN2.5-7B-INSTRUCT FINETUNING DETAILS AND COMPARISON WITH CDD

To contaminate Qwen2.5-32B-Instruct on MMLU we selected 10,000 samples from the train set of cais/mmlu on the huggingface hub as well as 100 samples from the test set that we duplicated 100 times (split A). The split B is made of 100 other samples from the test set that are not included in the training set. Finetuning is performed using QLoRA Dettmers et al. (2023) instead of full finetuning, as we did in the other experiments, because this LLM is a lot larger, and this finetuning was performed using 1 H100 and lasted approximately 8 hours for each model. The pipeline and hyperparameters are the default from alpaca Taori et al. (2023) explained in Appendix A with 5 epochs instead of 3. We tested different learning rates. The main paper shows results for a learning rate of 5e-5 in Table 2. Figure 5 shows the same results in a Figure. Figure 4 shows results for a learning rate of 5e-4 yielding very similar results (see Table 4 for metrics), and Figure 6 for a learning rate of 5e-6. In this last figure, the learning rate is very small which doesn't change very much the way the LLM is generating the question text. As such, LogProber has a hard time noticing contamination (as it is very light on the question). However, CDD seems to maintain its accuracy on the coding benchmark. While this is an unexpected behavior we believe it could come from the fact that when predicting the question, for the first few tokens the LLM doesn't have enough context to predict them accurately, this naturally increases the Safe Score of LogProber as the cumulative logprobability curse is less horizontal. However, the answer being generated after the question, the LLM has enough context to notice it knows the generation by heart which may lead it to retrieve answers more easily than questions.

Table 4: **Contamination experiment metrics in QA setting**.

| Algorithm | Dataset | Learning Rate | Accuracy | Precision | Recall | F1 Score |
|-----------|---------|---------------|----------|-----------|--------|----------|
| CDD       | MMLU    | 5e-4          | 0.50     | 0.50      | 1.0    | 0.67     |
| LogProber | MMLU    | 5e-4          | 1.0      | 1.0       | 1.0    | 1.0      |
| CDD       | MMLU    | 5e-5          | 0.50     | 0.50      | 1.0    | 0.67     |
| LogProber | MMLU    | 5e-5          | 0.98     | 1.0       | 0.95   | 0.97     |
| CDD       | MMLU    | 5e-6          | 0.5      | 0.5       | 1.0    | 0.67     |
| LogProber | MMLU    | 5e-6          | 0.57     | 1.0       | 0.13   | 0.23     |
| CDD       | CODE    | 5e-4          | 0.63     | 0.58      | 0.95   | 0.72     |
| LogProber | CODE    | 5e-4          | 1.0      | 1.0       | 1.0    | 1.0      |
| CDD       | CODE    | 5e-5          | 0.68     | 0.62      | 0.93   | 0.74     |
| LogProber | CODE    | 5e-5          | 0.99     | 1.0       | 0.99   | 0.99     |
| CDD       | CODE    | 5e-6          | 0.76     | 0.68      | 0.97   | 0.80     |
| LogProber | CODE    | 5e-6          | 0.5      | 1.0       | 0.01   | 0.02     |

A similar setup was made for the coding benchmark with the first file of MBXP Athiwaratkun et al. (2023) being used to extract 10,000 samples for the training set and then 100 samples were taken from the bigcode/bigcodebench dataset for the test set. Similarly, they were duplicated 100 times each for the training set for split A, and another 100 were used to compose split B (which are not added to the training data).

For the STD training in MMLU we did only include the training set without splits A nor B making 10,000 datapoints in total. These were extracted from the question column for MMLU and we appended the answer column right after. For prompting reasons, because the question finishes with the "Answer:" prompt we added a left parenthesis afterwards to force the LLM to complete with the answer right after without reasoning. As the answer column starts with a left parenthesis we removed it to avoid having two of them. For the QA finetuning we included split A making 20,000 datapoints in total. For MBXP we used the content column for the train set and cut completions to the 1000th token in case they exceeded this length. For the splits A and B we used the complete_problem and canonical_solution columns appended after each other.

In order to evaluate CDD we need to stop the generation at some point to define the limit of what is an answer to a given question. While some LLMs generate an end of text token by themselves it is not always the case and we noticed finetuned Qwen models did not always put one and started hallucinating after the answer. Thus, we added a prompt asking the model to stop after the answer, and,

because it was not very effective in MMLU, we cut the answers to one character for this benchmark (the model generates the answer in the first token of the completion). For BigCodeBench we did not restrict the answers and let the model generate up to 100 new tokens.

CDD hyperparameters were taken from the original paper Dong et al. (2024): $\alpha$ is 0.05, $\xi$ is 0.01 and 50 answers were generated for each question with a temperature of 1 in addition to the greedily decoded answer.

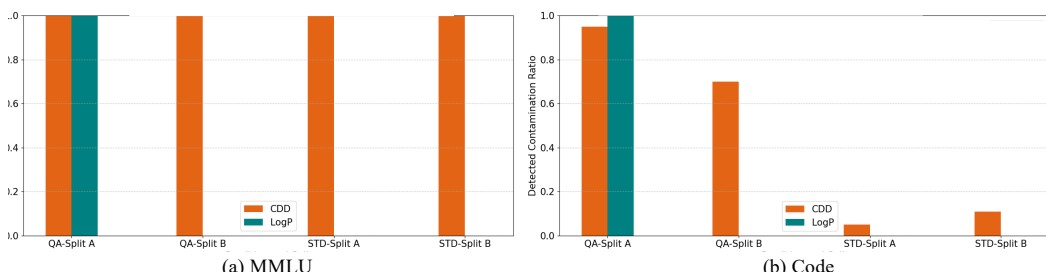

(a) MMLU        (b) Code

Figure 4: LogProber and CDD results on MMLU and BigCodeBench for learning rate 5e-4. These plots represent the suspected contamination ratio for CDD and LogProber on each split for two LLMs and two datasets. The left side corresponds to MMLU and the right side is BigCodeBench. For each dataset, the two left bars represents the contamination ratios for Qwen2.5-7B-Instruct finetuned on a training set and the split A in the QA format (trained to predict both the question and the answer). Split B is independant from A and is not used to train the models. The right two bars show results for a control with Qwen2.5-7B-Instruct finetuned on the same training set without split A nor split B. An ideal contamination detection algorithm is at 1 on the very left plot (QA-Split A) and at 0 for the other bars.

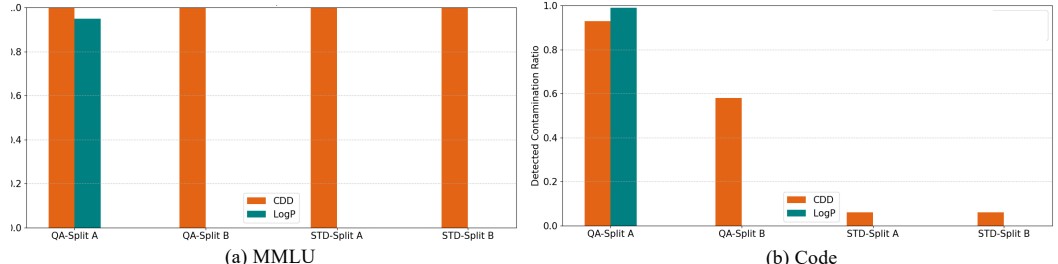

(a) MMLU        (b) Code

Figure 5: LogProber and CDD results on MMLU and BigCodeBench for learning rate 5e-5. These plots represent the suspected contamination ratio for CDD and LogProber on each split for two LLMs and two datasets. The left side corresponds to MMLU and the right side to BigCodeBench. For each dataset, the two left bars represents the contamination ratios for Qwen2.5-7B-Instruct finetuned on a training set and the split A in the QA format (trained to predict both the question and the answer). Split B is independent from A and is not used to train the models. with Qwen2.5-7B-Instruct finetuned on the same training set without split A nor split B. An ideal contamination detection algorithm is at 1 on the very left plot (QA-Split A) and at 0 for the other bars.

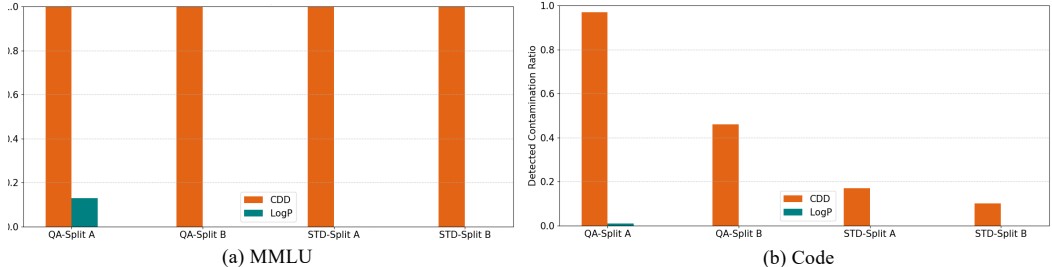

(a) MMLU        (b) Code

Figure 6: LogProber and CDD results on MMLU and BigCodeBench for learning rate 5e-6. These plots represent the suspected contamination ratio for CDD and LogProber on each split for two LLMs and two datasets. The left side corresponds to MMLU and the right side is BigCodeBench. For each dataset, the two left bars represents the contamination ratios for Qwen2.5-7B-Instruct finetuned on a training set and the split A in the QA format (trained to predict both the question and the answer). Split B is independant from A and is not used to train the models. The right two bars show results for a control with Qwen2.5-7B-Instruct finetuned on the same training set without split A nor split B. An ideal contamination detection algorithm is at 1 on the very left plot (QA-Split A) and at 0 for the other bars.

