# OpenReview forum: "LogProber: Disentangling confidence from contamination in LLM responses"
_ICLR.cc/2026/Conference — Submitted to ICLR 2026_

### Official Review · Reviewer_5GiR · 2025-10-28

**Soundness:** 3
**Presentation:** 2
**Contribution:** 2
**Rating:** 4
**Confidence:** 3

**Summary:**

This paper proposes a method, LogProber, to detect data contamination in LLMs. (1) The paper argues contamination cannot be detected reliably from local features such as a few high-probability tokens, since those may arise naturally from linguistic or contextual predictability. Instead, this paper frames contamination as a global property of the model’s familiarity with an entire sequence. It computes a cumulative log-probability curve over the question and then integrates the sorted values into a single 'safe score'. (2) Instead of analyzing answers, LogProber examines how confidently the model predicts the question text itself. (3) Experiments on CRT, MMLU, and BigCodeBench show LogProber effectively distinguishes contaminated and outperforms CDD in avoiding false positives.

**Strengths:**

* Simple, efficient, and suitable for black-box LLMs (only one forward pass).
* Shifts perspective from local token-level confidence to a global measure of sequence familiarity
* Offers a complementary view to existing answer-based contamination detection methods.

**Weaknesses:**

The main issue of this paper is its limited experimental coverage. The experiments are restricted to CRT, MMLU, and BigCodeBench, and only two model families are tested (llama 7B and qwen2.5 32b). For a method aimed at general-purpose contamination detection, it would be much stronger to evaluate across a wider variety of models and benchmarks (for example, standard QA tasks or additional reasoning datasets). Moreover, the comparison is limited to CDD. There are many other relevant approaches, such as exposure-based memorization metrics, or masked-completion probes. Including several additional black-box baselines would clarify where LogProber stands within the broader landscape.

**Questions:**

(1) This paper sorts the log-probabilities before integration. This makes the safe score stable but removes positional and contextual information. Early high-confidence tokens could be more diagnostic of contamination than late ones, yet this distinction is lost. Future work could explore weighting earlier tokens or measuring how early the cumulative curve saturates to capture “recognition onset” without losing robustness.

(2) There are several minor typos and formatting issues. (a) line 117 “adn” (b) inconsistent quotation marks, eg line 126 “A”

---

> ### Author Response · Authors · 2025-11-26
> **Author response**
>
> We thank the reviewer for their accurate comments. The authors are not available to implement these suggestions during the discussion period but will update the future version of the paper accordingly.
>
> Regarding the questions:
>
> 1. This is indeed a great suggestion. We agree with the fact that early tokens should be more informative about contamination than late ones and have experimented around that concept but haven't figured out an efficient way to implement it yet.
>
> 2. We thank the reviewer for pointing these out. We corrected them.

---

### Official Review · Reviewer_amUu · 2025-10-28

**Soundness:** 1
**Presentation:** 2
**Contribution:** 1
**Rating:** 2
**Confidence:** 4

**Summary:**

The paper addresses the issue of contamination in large language models, via a logprober. They propose to check the question of the QA, rather than the answer to avoid false positives where the model actually is smart and knows the answer. They use the cumulative log-probability to devise a Safescore. Experiments are done to show the effectiveness of their approach on CRT / Llama1, and Qwen2.5-32B.

**Strengths:**

1. Important problem, data contamination is still a difficult and important problem to be solved.
2. The paper explained their key ideas very clearly

**Weaknesses:**

1. Lack of innovation, there have been a wide range of confidence/logP based-scores  [1], and people have already figured that rephrasing would escape those detection methods [2, 3]. This method lacks merit in advancing the field.
2. Model / dataset used are too simple. Only one set of experiments are done (CRT / Llama-1) to show effectiveness.
3. Lack of analysis. Does question length play an effect here? What about CoT models?


[1] Zhang, Huixuan, Yun Lin, and Xiaojun Wan. "Pacost: Paired confidence significance testing for benchmark contamination detection in large language models." arXiv preprint arXiv:2406.18326 (2024).

[2] Yang, Shuo, et al. "Rethinking benchmark and contamination for language models with rephrased samples." arXiv preprint arXiv:2311.04850 (2023).

[3] Yao, Feng, et al. "Data contamination can cross language barriers." arXiv preprint arXiv:2406.13236 (2024).

**Questions:**

Minor concerns include:
1. Writing isn't polished through out the paper, many details are lacking: ’Q-A’ -> 'Q-A' (quotes), Confidence refers to a property of the
model’s response to a given input — namely, that the model’s answer is confident (AI written)
2. Citation format is incorrect, in-context learning abilities Brown et al. (2020a) -> in-context learning abilities (Brown et al. 2020a), ...

---

> ### Author Response · Authors · 2025-11-26
> **Author response**
>
> We thank the reviewer for their relevant feedback. The authors are not available to implement these suggestions during the discussion period but will update the future version of the paper accordingly.
>
> For the questions:
>
> 1. We thank the reviewer for pointing out that we forgot quotation marks around "Q-A" twice throughout the paper. However, we did not use AI to write our paper. The sentence "Confidence refers to a property of the model's response to a given input — namely, that the model's answer is confident" could indeed be improved to "Confidence refers to a property of the model's response to a given input — namely, that the model's answer is consistent" for greater clarity. We apologize for any confusion regarding the term "confidence."
>
> 2. We have updated the citation style and thank the reviewer for pointing out this inconsistency.

---

### Official Review · Reviewer_zdDS · 2025-10-31

**Soundness:** 3
**Presentation:** 3
**Contribution:** 2
**Rating:** 4
**Confidence:** 4

**Summary:**

The paper introduces "LogProber," a novel algorithm designed to detect data contamination in Large Language Models (LLMs) by disentangling "contamination" from "confidence". The authors argue that existing methods, which often focus on the model's answer (A), can confuse a model's high confidence (due to strong generalization) with contamination (due to memorization) .

**Strengths:**

The paper addresses a fundamental, high-impact problem. As models become more powerful, their performance on standard benchmarks is increasingly scrutinized for contamination . This work provides a practical tool to help maintain the integrity of LLM evaluation.

**Weaknesses:**

- The paper introduces a specific, non-trivial formula for the "Safe Score" based on the integral of the sorted cumulative log-probabilities (Equation 1). However, there is no justification provided for why this specific formulation is optimal, or even necessary, compared to simpler, more direct statistical measures of the "plateness" of the $log(p)$ curve. For instance, what about the simple variance of the $log(p)$ values? Or the 10th percentile of $log(p)$? A contaminated sequence should have many $log(p) \approx 0$, which would heavily skew these simpler metrics. The paper asserts this metric works, but does not provide an ablation study or theoretical justification for this specific choice over more intuitive alternatives.

- The paper's motivation is rooted in detecting contamination within the "gargantuan... corpora" used for pretraining. However, all the controlled experiments simulate contamination by fine-tuning an existing model , sometimes with significant repetition (e.g., seeing each item 30 times ). This is a very strong, explicit "memorization" signal. It is not self-evident that this signal is representative of true pretraining contamination, where a benchmark item might be seen only a handful of times within a multi-trillion token corpus. The experiments clearly show LogProber detects fine-tuning contamination, but its sensitivity to the (likely weaker) signal of pretraining contamination remains unproven.

- The paper repeatedly uses a fixed threshold of safe_score < 1 to flag contamination. This seems practically brittle. It is highly plausible that the baseline Safe Score for uncontaminated text is model-dependent. For example, a larger, more capable model (e.g., Llama-70B) will inherently find most text less "surprising" than a smaller model (e.g., Llama-7B), which would likely result in a lower baseline Safe Score, even without contamination. Asserting a single universal threshold without exploring its scaling properties across different model sizes and families is a significant oversimplification. This suggests a model-specific calibration of the "clean" baseline score might be necessary, undermining the method's simplicity.

- This framework also fails to account for contamination beyond the token level, such as contamination at the semantic level. There are doubts about whether this can truly enhance the integrity of the LLM evaluation.

**Questions:**

1. Could the authors provide justification for the specific "Safe Score" formulation in Equation 1? What simpler statistical metrics were considered (e.g., variance of $log(p)$, skewness, or the ratio of tokens with $log(p) > -0.1$) and why was this complex integral-based approach chosen over them?

2. The paper proposes a universal threshold of safe_score < 1. How does this threshold hold across different model scales (e.g., 7B vs. 70B vs. 180B) and architectures (e.g., Qwen vs. Llama vs. Mistral)? Does this threshold need to be re-calibrated for each model, and if so, how does that affect the method's practical utility?

3. The experiments use fine-tuning to simulate contamination . How confident are the authors that this signal is a valid proxy for pretraining contamination , which is arguably a much weaker signal (e.g., a single pass over a test item) than the multi-epoch fine-tuning performed?

---

> ### Author Response · Authors · 2025-11-26
> **Author response**
>
> We thank the reviewer for their useful feedback. The authors are not available to implement these suggestions during the discussion period but will update the future version of the paper accordingly.
> Regarding the questions:
>
> 1. The Safe Score formula may appear complex, but it is based on a simple observation from Figure 1: contaminated sequences exhibit a distinctive cumulative log-probability curve. We designed the formula to quantify this curvature. While alternative approaches exist—such as analyzing log-probability distributions or counting low-probability tokens—these methods have limitations. Simple statistics like variance are dominated by low-probability tokens (high logp) and fail to adequately capture the behavior of high-probability tokens (low logp - where contamination signal lies). Additionally, threshold-based methods require arbitrary cutoff values that are difficult to justify. Our Safe Score addresses these issues by using a continuous measure (the area under the curve) that is high when cumulative log-probability increases smoothly (no high probability token) and low when the increase is irregular (presence of high probability tokens).
>
> 2. We did not particularly investigate the importance of the threshold value across model scale and architecture. This is a good suggestion to improve the validity of the method. Nonetheless, most models nowadays have a very similar architecture (except for the hidden size and number of layers for larger models).
>
> 3. The reviewer raises a valid point. It is computationally unfeasible to pretrain an LLM to test real pretraining contamination. The finetuning contamination we explore in this paper tries to mimic pretraining contamination in the most similar setup as possible with a full language modeling objective. We show the algorithm works well to detect contamination with this form of training; it is reasonable to extrapolate that it would scale to pretraining, but it is difficult to prove in practice without investing hundreds of thousands of dollars in pretraining an LLM.

---

### Official Review · Reviewer_AyQU · 2025-11-01

**Soundness:** 2
**Presentation:** 2
**Contribution:** 2
**Rating:** 2
**Confidence:** 3

**Summary:**

This paper introduces LogProber, an algorithm to detect data contamination by disentangling a model's true confidence from its memorization.


The key problem LogProber solves is that existing methods (like CDD) mistake high confidence for contamination . On tasks a model is good at (like MMLU), these methods falsely flag all answers as "contaminated" .



LogProber's solution is to analyze the question text instead of the answer . It calculates a "Safe Score" by measuring if the question's text is "familiar" (contaminated) or "surprising" (clean) to the model

**Strengths:**

1. Solves the "Confidence" Flaw: The paper's main strength is identifying that existing detectors mistake a model's high confidence for contamination. Its novel solution is to analyze the question text instead of the answer, successfully disentangling genuine skill from memorization.


2. High Transparency: The authors are rigorous and transparent about the tool's limitations. They explicitly demonstrate that LogProber is blind to "answer-only" (-A) contamination, which is a common format for fine-tuning, making the tool's precise capabilities clear.

**Weaknesses:**

1. The writing can be improved -- the introductory content is too long and the citation format can be further improved. Most importantly, the paper will benefit from adding a conclusion section and related work section. These two are clearly missing.

2. The Llama-1-7B model used in the experiment is too old. And the baselines are too few and not strong enough. It only compared with CDD, while data contamination detection is not a new topic and there are a lot of existing work defining and addressing this problem like Shared Likelihood[1], Guided Prompting, N-Gram Accuracy, and Choice Confusion.

3. The LogProber algorithm detects contamination involving *questions* (Q-type) but fails for *answer-only* (A-type) contamination, common in fine-tuning stages.

[1] Proving test set contamination for black-box language models. ICLR 2023

[2] Time travel in llms: Tracing data contamination in large language models. ICLR 2024

[3] Benchmarking benchmark leakage in large language models. ArXiv

[4] Data Contamination Can Cross Language Barriers. EMNLP 2024

**Questions:**

1. How does the algorithm get the semantic-equivalent sequence of origin question? If it's also generated by the target LLM, the correctness can't be guaranteed.

2. Is there any theoretical prove of the 1 boundary of safe score? Otherwise the Safe Score cutoff (<1) lacks theoretical justification or adaptive calibration.

3. How will your method perform on some open-source / private LLMs? Will there be any contamination detected?

---

> ### Author Response · Authors · 2025-11-26
> **Author response**
>
> We thank the reviewer for their thoughtful feedback. The authors are not available to implement these suggestions during the discussion period but will update the future version of the paper accordingly.
>
> As for the questions:
>
> 1) I am not entirely sure to understand the question. The algorithm doesn't use any semantic-equivalent sequence of the origin question. It simply runs on a given sequence tested for contamination and doesn't generate any semantic-equivalent version of it.
>
> 2) This value has been empirically selected akin to most other algorithms in the field. There is no mathematical reason for 1 to be best. It was the value that seemed to work best in our experiments.
>
> 3) We tested it on Llama 1 on the CRT questionnaire and the model accurately found contamination on the items proportionally to their presence in the litterature: items 1,2,3 are very famous and reflect high levels of contamination, items 4,5,6,7 are less present in the litterature and present significant but weaker scores of contamination. On reframed version of these items they present no contamination.

---

### Meta-Review · Area_Chair_gwSb · 2025-12-21

**Summary:**

Reviewers were concerned with the methodological novelty, justification (and intuition) of the proposed method, and lack of empirical evidence, such as only showing few datasets and model families, weak baselines, and contamination setup.

**Reviewer Concerns:**

The authors note they cannot implement suggested improvements during the short discussion period, so the rebuttal mainly acknowledged limitations and provided clarifications.

**Reviewer Scores:**

Since most of the empirical suggestions from reviewers remain, I believe most reviewers would have maintained their score.

---

### Decision · Program_Chairs · 2026-01-26

Reject